# Flexible Machine Learning Algorithms for Clinical Gait Assessment Tools

**DOI:** 10.3390/s22134957

**Published:** 2022-06-30

**Authors:** Christian Greve, Hobey Tam, Manfred Grabherr, Aditya Ramesh, Bart Scheerder, Juha M. Hijmans

**Affiliations:** 1Department of Rehabilitation Medicine, University Medical Center Groningen, University of Groningen, 9713 GZ Groningen, The Netherlands; j.m.hijmans@umcg.nl; 2Department of Human Movement Sciences, University Medical Center Groningen, University of Groningen, 9713 GZ Groningen, The Netherlands; 3Oro Muscles B.V., 9715 CJ Groningen, The Netherlands; ht@tardigradellc.com (H.T.); manfredgrabherr@gmail.com (M.G.); 4Department of Medical Biochemistry and Microbiology, Uppsala University, 751 23 Uppsala, Sweden; 5Department of Biomedical Engineering, University Medical Center Groningen, University of Groningen, 9713 GZ Groningen, The Netherlands; a.ramesh.5@student.rug.nl; 6Center for Development and Innovation (CDI), University Medical Center Groningen, University of Groningen, 9713 GZ Groningen, The Netherlands; b.scheerder@umcg.nl; 7Data Science Center in Health (DASH), University Medical Center Groningen, University of Groningen, 9713 GZ Groningen, The Netherlands

**Keywords:** clinical gait analysis, gait partitioning, machine learning, wearables, inertial measurement units, sensors, deep neural networks, reinforcement learning

## Abstract

The current gold standard of gait diagnostics is dependent on large, expensive motion-capture laboratories and highly trained clinical and technical staff. Wearable sensor systems combined with machine learning may help to improve the accessibility of objective gait assessments in a broad clinical context. However, current algorithms lack flexibility and require large training datasets with tedious manual labelling of data. The current study tests the validity of a novel machine learning algorithm for automated gait partitioning of laboratory-based and sensor-based gait data. The developed artificial intelligence tool was used in patients with a central neurological lesion and severe gait impairments. To build the novel algorithm, 2% and 3% of the entire dataset (567 and 368 steps in total, respectively) were required for assessments with laboratory equipment and inertial measurement units. The mean errors of machine learning-based gait partitions were 0.021 s for the laboratory-based datasets and 0.034 s for the sensor-based datasets. Combining reinforcement learning with a deep neural network allows significant reduction in the size of the training datasets to <5%. The low number of required training data provides end-users with a high degree of flexibility. Non-experts can easily adjust the developed algorithm and modify the training library depending on the measurement system and clinical population.

## 1. Introduction

Walking is the most important form of mobility and allows us to participate in labour, societal and sports-related activities. To regain a healthy, normal walking pattern after injury or disease is, therefore, one of the most important goals in clinical rehabilitation [1,2,3]. Effective gait rehabilitation is predicated on accurate diagnostics. The current gold standard of gait diagnostics is instrumented, laboratory-based three-dimensional (3D) motion-capture analysis, including measures of external forces and muscle activity (electromyography) (3D clinical gait analysis (3D CGA)) [4,5]. Using 3D CGA, clinicians can quantify abnormalities in joint motions, joint loading and muscle activity, and design patient-specific interventions. While 3D CGA has improved treatment outcomes after rehabilitation [4,5,6,7], it has some practical disadvantages. For example, 3D CGA is dependent on large, expensive motion-capture laboratories and highly trained clinical and technical staff to guarantee quality during the data processing and interpretation steps. These requirements for staff and equipment constrain the use of objective gait diagnostics to a limited number of highly specialized hospitals. To improve the accessibility of objective gait diagnostics and provide more patients with targeted, personalized gait rehabilitation, less technically demanding and laboratory-independent solutions for gait diagnostics are needed.

An alternative to laboratory-based 3D CGA are wearable sensor systems coupled with machine learning analytics [8,9,10,11,12]. These sensor systems use predictive machine learning methods to automatically partition and analyse gait (e.g., foot-contact and foot-off events) from sensor signals (e.g., inertial measurement units (IMUs)) [8,9,10,11,12,13]. These sensor systems are mobile, cost-effective and allow the automation of some tasks that currently require laboratory equipment and skilled knowledge [8,9,10,11,12,13]. For example, partitioning gait into stance and swing phases is an important step in 3D CGA. Gait partitioning allows comparison with normative data and the formulation of a medical diagnosis. For example, partitioned recordings of muscle activity (e.g., from electromyographic sensors) allow the diagnosis of muscle spasticity in patients with a central neurological lesion [4,5,6,7].

The current gold standard for gait partitioning uses laboratory-based force plate data to detect foot-contact and foot-off events [14]. While force-plate-based gait partitioning is very accurate, it (1) requires expensive and stationary equipment, (2) does not usually allow partitioning of more than 2–3 steps within a trial, (3) significantly loses accuracy in severely impaired patients with very small step sizes or impaired foot clearances and (4) requires trained technical staff to manually control the partitioning accuracy and correct partitions if needed. Machine learning-based gait partitioning is a promising alternative because it can be performed with a single, wearable and cheap IMU sensor on the foot or pelvis, which allows the inclusion of more steps in the diagnostic process and can be performed in the patients’ home environments [8,9,10,11,12].

While machine learning-based sensor systems are promising to improve the accessibility of CGA, they still have some limitations. One main limitation is the lack of technical flexibility (e.g., dependence on measurement systems) and clinical flexibility (e.g., dependence on specific patient groups). For example, the majority of currently used machine learning-based sensor systems were validated in healthy adults or built for rather narrow groups of patients with similar gait impairments [8,11]. Once a newly assessed patient differs too much from the original patient population, these algorithms significantly lose accuracy and become invalid. This lack of clinical flexibility was addressed by Kidzinski et al. (2019) by making use of large training datasets of different walking patterns [15]. A total of 9092 annotated 3D CGA recordings (80% of the total dataset) were required to accurately identify swing and stance phases with artificial neural networks (long–short-term memory) [15]. While the partitioning accuracy of the algorithm was good, the dependency on large training datasets limited its technical flexibility. For example, an entirely new training dataset would be required to re-build the algorithm from Kidzinski et al. (2019) for use with an IMU sensor system or if one would like to add other gait features than foot-contact and foot-off events to the analysis. Extensive re-training procedures are infeasible and often impossible to perform for non-experts and less specialized hospitals, local clinics or physiotherapy departments.

We address the current limitations in machine learning-based gait assessment tools by developing a highly flexible machine learning method that requires only a few training datasets and can be easily modified by end-users without the need for laboratory-based ground truth data sources. The proposed solution consists of a novel, reinforced deep neural network and a dynamic, living training library that can change and adapt over time with end-user input (Oro Muscles B.V, Groningen, The Netherlands). Contrary to previous attempts, the newly designed algorithm recognizes patterns in motion signals (e.g., accelerometer data) and identifies gait features, such as foot-contact and foot-off events, through reinforcement, rather than statistics alone. We hypothesize that, by giving more autonomy to end-users through reinforcement learning, our proposed approach results in an order of magnitude less training data, as well as orders of magnitude more technical and clinical flexibility in applicable use cases.

We will first establish the flexibility and accuracy of the reinforced deep neural network for laboratory-based 3D CGA patient recordings and then in a wearable IMU sensor system. Finally, we present a clinical use case for the AI tool and IMU sensor system by visualizing AI-based partitions of the electromyographic (EMG) signal of the gastrocnemius muscle in patients with a central neurological lesion. Time-normalized visualizations of gastrocnemius muscle activity are a frequently used method to identify abnormalities in calf muscle activity and diagnose spasticity [4,5,6,7].

## 2. Materials and Methods

To address the main aim of the current study, we used a two-way approach. First, accelerometer data from 3D CGA patient recordings were used to train the AI tool for laboratory-based assessments. In the second step, the AI tool was trained and validated with IMU-based accelerometer data from the Oro Muscles IMU sensor system.

### 2.1. Algorithm Development for Laboratory-Based 3D CGA Data

#### 2.1.1. 3D CGA Patient Recordings

Historical data recordings between April 2021 and June 2021 were selected from the database of the motion laboratory of the University Medical Center Groningen, The Netherlands. Datasets were included if patients previously signed informed consent, were diagnosed with a central neurological lesion and full 3D CGA data were acquired. The patient recordings consisted of on average eight-meter walking trials during regular clinical visits at the motion laboratory of the UMCG Groningen, The Netherlands. In total, 60 walking trials from 14 patients were included in the analysis. The right and left legs were treated separately, resulting in a total of 120 datasets.

#### 2.1.2. 3D CGA Data Processing and Spatio-Temporal Parameter Computation

The 3D CGA patient recordings consisted of 3D marker position data of the plug-in gait model (2010) recorded at 100 Hz with 10 optical cameras (Vero) and Vicon motion capture software Nexus 2.12. Two Amti force plates recorded the ground reaction force data at 1000 Hz during gait recordings.

The 3D CGA-based identification of foot-contact and foot-off events was based on the in-built Vicon Nexus algorithm using force plate data. Initial foot contact was detected once vertical ground reaction forces exceeded 10 Newton. The moment of foot-off was defined once the vertical ground reaction force decreased below 10 Newton. All gait events were manually checked by an experienced lab technician and corrected if needed. In cases where force plate signals were corrupted because the gait deviated too much, gait events were manually annotated by an experienced lab technician and visually checked by the principal investigator of this study.

Step lengths were computed based on the absolute distance between the left and right lateral malleolus markers from the plug-in gait 3D CGA model. The 3D CGA-based foot-contact events were used to define the moment of maximum step length. Gait speed was computed based on the absolute distance (m) travelled by the anterior superior spine marker divided by time (s). All spatio-temporal gait data were averaged across steps within a trial and across trials of the same condition (barefoot and shoes/orthotics) per participant.

The total number of steps made within a trial was computed from 3D ankle marker position data. First, the difference between the anterior–posterior left and right lateral malleolus marker positions was computed for each 3D CGA trial. In the next step, the number of peaks in the left–right ankle distance signal was used to count the number of steps per trial. The sum of all steps across trials was used as the total number of steps for each condition and participant. Custom Python scripts (v. 3.8) were used to compute the spatio-temporal parameters and total number of steps. The accuracy of the step detection algorithm was visually checked by the principal investigator and corrected if needed. 

#### 2.1.3. Implementation, Training and Accuracy Testing of the AI Tool

To build the AI tool, an existing ML algorithm, Saguaro, consisting of a Hidden Markov Model, self-organizing map and a generative module [16], was combined with end-user feedback through a graphical user interface (GUI) and reinforcement learning. The original self-organizing map was replaced with a proprietary generative deep learning network (fuzzy logic algorithm) (Oro Muscles BV, Groningen, The Netherlands). The Oro Muscles deep learning network allowed the accommodation of user inputs for unsupervised gait partitioning. The implementation of the newly developed ML algorithm and training workflow is depicted in Figure 1 with sample user interfaces and is explained in more detail below.

First, the recorded EMG and IMU signals were pre-processed with a smoothing algorithm (Fast Fourier Transform (FFT) and Zurbenko–Kolmogorov Filter (ZK)) [17]. The processed EMG and IMU signals were then fed into a Saguaro-like unsupervised algorithm to segment the data into stance and swing phases (unsupervised learning). Next, the partitioned results were displayed in a GUI and corrected by the user if needed. The end-user manually selected the signal pattern of interest from the GUI (e.g., start and end of the acceleration signal of the swing phase). During this step, the user was guided by the laboratory-based gait partitions. After the end-user selection of the correct patterns, the final start and end coordinates of the swing phase partition were fed back to the ML algorithm in a reinforced learning loop. A custom-made java script was implemented in the AI tool’s user interface to allow the transfer of the end-user inputs to other datasets. 

In the current study, the AI tool was trained with manually annotated hints between foot-contact and foot-off events from the mtp-2 marker acceleration signal (Figure 2 and Appendix A). In a next step, a fuzzy logic algorithm in the time domain was applied to match the selected interval against any subset of the recording. This step produced interval instances, or “cycles”, where overlaps between these cycles were restricted. To ensure that the cycles of motion (e.g., swing phases) did not coincide, a sliding window (Hamming distance) was used. This procedure was repeated iteratively until the algorithm converged and found a discrete set of cycles (Figure 2).

The force plate data and manual partitions from the 3D CGA system were used as the ground truth data for assessing the AI tool’s partitioning accuracy. Partitioning accuracy was determined by computing the mean difference between foot-contact and foot-off events from the ground truth with the AI tool’s partitions. If the partitioning error exceeded 0.060 s, the corresponding data set was transitioned into the training set and another hint was created.

Custom MATLAB (R2021a Mathworks, Newark, DE, USA) and Python (v. 3.8) scripts were used for data processing, analysis, feature extraction and computation of spatio-temporal gait variables.

### 2.2. Experimental Validation with IMU System and Clinical Use Case

#### Experimental Procedure and Data Collection

The first five consecutive patients visiting the motion laboratory between June 2021 and December 2021 eligible for inclusion were asked to participate in the IMU-based validation study of the AI tool. Next to the plug-in gait marker set, Cometa EMG sensors were placed according to the SENIAM guidelines on all major superficial lower limb muscle groups (vatus medialis, rectus femoris, semitendinosus, medial head of the gastrocnemius, soleus and tibialis anterior muscle). EMG data were recorded at 1000 Hz as part of the usual clinical 3D CGA. In addition, two Oro Muscle IMUs and EMG sensors were placed at the shank and foot and gastrocenemius and tibialis anterior muscles, respectively (Figure 3). The Oro Muscle’s EMG sensors were placed proximal to the Cometa EMG sensors. The multiple channels of the Oro sensor system were connected together via a raspberry pi for data acquisition. Oro EMG and IMU data were recorded at sample rates of 500 hz and 100 hz, respectively. The Oro Muscle sensor data were time-synchronized with the 3D CGA-based data through post-processing by syncing the peaks of the accelerometer data of each step of the patient through a custom MATLAB script.

3D CGA-based event detection was the same as described in 2.1. Cometa and Oro Muscles EMG data were bandpass-filtered between 20 (high band) and 450 Hz (low band) with a fourth-order Butterworth filter. To create the linear envelope, EMG data were rectified and low-pass-filtered at 10 Hz with a fourth-order Butterworth filter.

## 3. Results

### 3.1. Laboratory-Based 3D CGA

#### 3.1.1. Training and Test Datasets

In total, five steps and six hints from four patients were fed into the AI tool for algorithm training. Table 1 gives the spatio-temporal gait parameters of the trial included in the training dataset (Table 1). Each row denotes data from one participant and the corresponding condition (barefoot and shoes/orthotics). The trials used for training were excluded from the test set. The 3D CGA test dataset consisted of a total of 567 steps (291 right and 276 left steps) from 13 different patients (eight children (12.8 ± 3 years) and five adults (43.8 ± 14.6 years)) with severe walking dysfunctions (Table 2). Hence, less than 2% of the total number of included steps was used for algorithm training. From one patient, two datasets before and after the treatment of muscle spasticity were included in the validation dataset. Nine of the included patients were diagnosed with spastic cerebral palsy, one patient with dystonic cerebral palsy, one with an incomplete spinal cord lesion, one patient with an unknown lesion of the central nervous system and one with a primary lateral sclerosis.

#### 3.1.2. Partitioning Accuracy

When compared with the ground truth data, the absolute average difference in foot-off and foot-contact event detection was 0.021 s (±0.021 s). The maximum time difference between laboratory-based and machine learning-based event detection was 0.08 s.

### 3.2. Wearable Sensor-Based CGA

#### 3.2.1. Participant Characteristics

In total, five patients participated in the experimental validation study with the wearable sensor system. Table 3 gives the average spatio-temporal gait parameters of the included participants. On average, the patients were 59.2 years old (±14.6) and had a BMI of 27.2 (±4.5). Two participants were diagnosed with a stroke, one with cerebral palsy, one with an incomplete spinal cord injury and one patient with multiple sclerosis. Table 3 provides an overview of the basic gait parameters of the included patients and the corresponding barefoot and shoe conditions.

#### 3.2.2. Datasets and Algorithm Training

For the partitioning of the IMU sensor system data, one step from every walking condition and each patient (barefoot, with shoes/orthotics) was used as a hint (nine hints in total). Hence, to re-train the AI tool and allow the partitioning of all 368 steps in swing and stance phases, less than 3% of the entire dataset was required for re-training. 

#### 3.2.3. Partitioning Accuracy

When compared with laboratory-based 3D CGA partitioning, the average 3D CGA-based AI partitioning error was 0.035 ± 0.028 s and the sensor-based AI partitioning error was 0.032 ± 0.024 s.

#### 3.2.4. Clinical Use Case

Figure 4 shows an example case of a visualization of the AI tool-based EMG partitioning of the gastrocnemius muscle activity into stance and swing phases. Figures in the Appendix A provide all EMG and accelerometer partitions of the included participants.

## 4. Discussion

Combining a Saguaro algorithm with end-user-informed reinforcement learning and deep neural networks allows successful gait partitioning of 3D CGA data recordings as well as wearable, sensor-based IMU recordings. More importantly only a few manually annotated training hints were required to achieve accurate partitions. Only 2% (six hints out of 567 steps) of the entire dataset was required to accurately partition the 3D CGA recordings in severely impaired patients with a range of different gait impairments (Table 2). For sensor-based partitioning, about 3% (9 hints out of 368 steps in total) was required to achieve accurate partitioning of swing and stance phases. In comparison, previously used machine learning models required up to 80% of the entire dataset (>9000 datasets) [15] to achieve a similar level of partitioning accuracy or were confined to healthy subjects or small groups of rather similarly walking patients [8,11]. Therefore, the proposed ML workflow provides a new methodology to significantly reduce the burden associated with algorithm training as compared with previous ML-based methods for gait partitioning [8,15].

There is one main difference in ours as compared with current machine learning approaches for gait partitioning. By incorporating reinforcement learning and a deep neural network, the current AI tool does not try to discern patterns in the signals alone, but is aided by end-users to select patterns of interest. While this reinforcement does require more user input, it reduces the required training data in orders of magnitude and yields a high degree of technical and clinical flexibility. Only a few, manually annotated reinforcement hints are required to adjust the deep neural network and search for new patterns in the signal. These features allow clinicians to easily adjust the AI tool to different measurement systems (e.g., laboratory vs. IMU systems), analysis of different output signals (e.g., acceleration vs. velocity signals) or locomotor activities (e.g., walking vs. running). This clinical and technical flexibility is an important aspect for facilitating the use of CGA in a broad clinical context and making it accessible for more patients and non-specialized clinics or physiotherapy practices.

Another important feature of the novel AI tool is that it does not require the exact time points from laboratory-based ground truth data sources (e.g., force-plate-based partitions) for training. Instead, the AI tool uses manual inputs from the GUI to identify relevant patterns in the signal (Oro Muscles, B.V.; Figure 1). Once the moment of foot contact and foot off is roughly known, the AI tool detects the exact start and end points of the stance and swing phases automatically through a deep neural network incorporating a fuzzy logic algorithm in the time domain. This implementation of a fuzzy logic algorithm in the training process allows end-users to easily modify any existing training library. Even though it was not tested in the current study, end-users might solely rely on time-synchronized video recordings to modify the training library.

Next to cost-effectiveness and usability, AI-based solutions have high potential to improve diagnostic validity, decrease personnel costs associated with CGA, improve the accessibility of CGA to non-specialized hospitals and physiotherapy practices and facilitate remote gait rehabilitation. The novel AI tool allowed us to include more than 10 times as many steps (18.4 steps on average) as traditional force-plate-based methods (1–2 steps on average) per patient. Including more steps in the diagnostic process improves the diagnostic validity, especially in young children and patients with complex movement disorders. In addition, automating gait partitioning can reduce the time required for data processing during traditional 3D CGA by about 30 min per trial [15], leading to high savings in costs and engineering time. Finally, the AI tool has the potential to facilitate remote gait rehabilitation by allowing the automatic partitioning of IMU signals, independent of the type of measurement system or output data.

While our results and previous studies showed that machine learning has the potential to replace force-plate-based gait partitioning, their implementation is scarce, and it remains challenging to automate diagnostic steps in the CGA process. We made a first step by visualizing the AI-based partitioned muscle activity of the gastrocnemius muscle (Figure 4 and Appendix A). Visualizing gastrocnemius muscle activity as a percentage of the gait cycle allows comparison with normative data and the identification of abnormalities in the timing of muscle activity and, hence, muscle spasticity. However, to formulate a distinct diagnosis of muscle spasticity, one would need additional information (e.g., time-synchronized knee and ankle angular velocity or gastrocnemius-muscle-lengthening velocities) [4,5,6,7]. Until now, this kinematic information can only be acquired from complex kinematic models or wearable sensor systems with multiple IMUs on the lower limb segments [18]. However, future studies need to establish whether incorporating kinematic models or multiple IMUs into AI-based systems is feasible.

We propose that AI-based solutions should not aim to replace laboratory-based 3D CGA systems and biomechanical interpretation steps, but should aim to provide non-specialized clinics with the ability to perform basic assessments or screening of key gait functions. For example, sensor-based gait assessment tools could be used to inform clinicians whether or not abnormalities in muscle activity are present, and further investigation with more advanced systems is required. Other feasible use cases for AI-based sensor solutions can be automating the assessment of abnormal foot positioning during stance, problems in foot clearance or deficiencies in spatio-temporal parameters (e.g., stride time variability or gait speed). Spatio-temporal parameters are especially relevant because they are an important metric of overall mobility and are often used to evaluate the effect of interventions [1,2,3].

The following aspects warrant consideration when using the AI tool in daily clinical care or for research purposes. When training the AI tool for laboratory-based 3D CGA accelerometer data, one needs to account for the polarity of the acceleration signal with respect to the walking direction. The change in polarity when changing walking direction in the laboratory doubles the number of hints required. Using input data that do not change polarity as a consequence of the walking direction (e.g., kinematics or IMU systems) could, therefore, half the number of required manually annotated hints.

Before actual use in clinics, potential end-users may need to re-train the AI tool based on their own patient population, measurement system and primary data source. While all algorithms are implemented in C++ and run on Linux, MacOS and Windows 10, without any dependencies on third-party packages, their actual implementation would require some technical expertise. The GUI for selecting swing intervals is written in Java and accesses the partitioning algorithm via the Java Native Interface (JNI). Finally, before implementation into daily clinical patient care, the user-friendliness of the AI tool should be improved and further validation steps in larger patient populations are required.

## 5. Conclusions

Combining reinforcement learning with a deep neural network allows significantly smaller training datasets (<5%) with comparable accuracy in gait partitioning for laboratory-based 3D CGA as well as wearable sensor systems. The low number of required training data and ease of training through the GUI provide end-users with the flexibility to use the AI tool in different measurement systems and clinical populations. Future studies will show the potential of the novel AI tool to allow automatic diagnostics of foot contact and spatio-temporal gait parameters, and facilitate the accessibility of CGA in a broad clinical context.

## Figures and Tables

**Figure 1 sensors-22-04957-f001:**
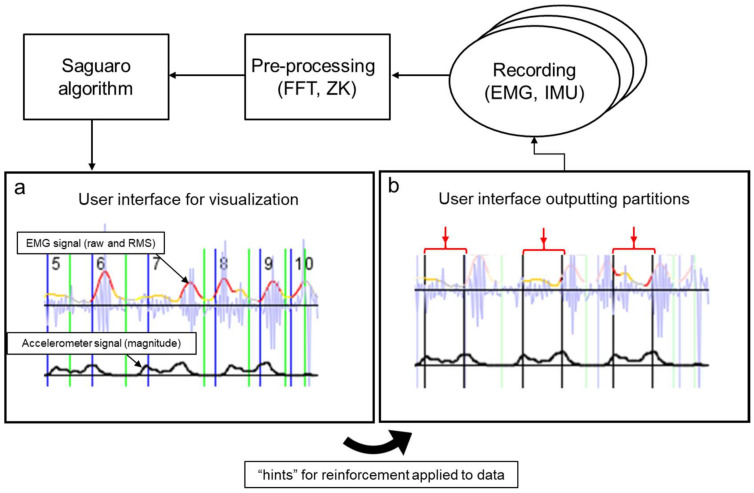
AI tool workflow for implementation and training utilizing the Saguaro algorithm, user feedback and reinforcement learning. Panel 1 (**a**) depicts the unsupervised partitions (5–10, green and blue vertical lines) from the Saguaro algorithm on the EMG and accelerometer signal. RMS = root-mean-square error. Panel 1 (**b**) depicts the final partitions after reinforcement learning on the EMG and the accelerometer signal (red arrows indicate swing phases).

**Figure 2 sensors-22-04957-f002:**
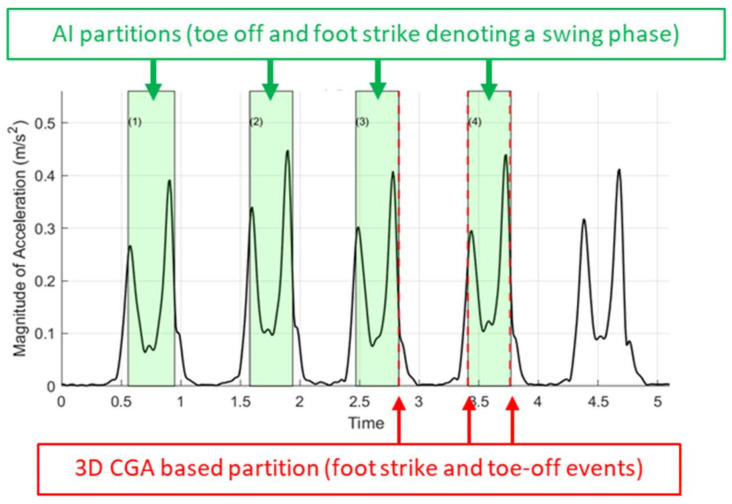
Sample IMU accelerometer signal and AI-based swing phase partitions (green shaded areas) of four out of five gait cycles. The user feedback from the GUI (Figure 1) was used to partition all datasets not included in training. Unshaded areas represent stance phases. Laboratory-based timings of foot-contact and foot-off events (red arrows) were used as ground truth to compute the partitioning error.

**Figure 3 sensors-22-04957-f003:**
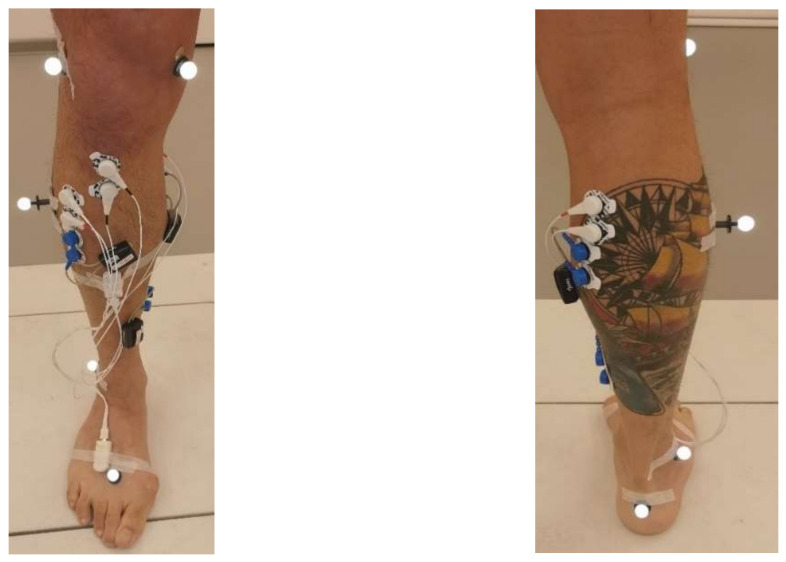
Oro muscles B.V. and 3D CGA sensor set-up.

**Figure 4 sensors-22-04957-f004:**
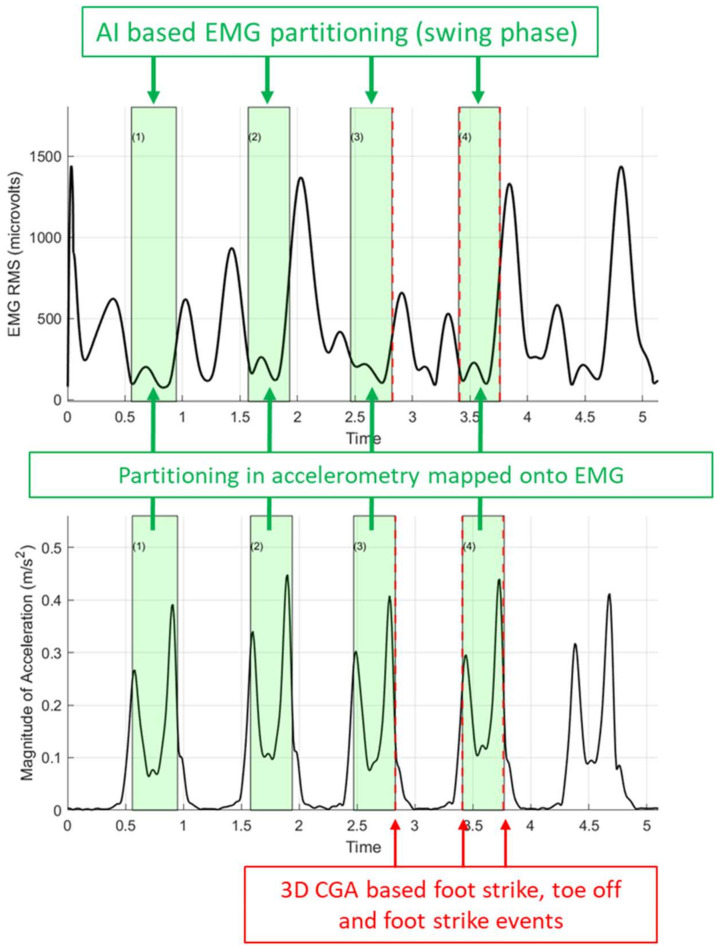
Representative sample of AI-based EMG partitioning of the right gastrocnemius muscle. The upper panel shows the Oro Muscles EMG envelope of a representative participant; the lower panel shows the accelerometer signal from the right IMU foot sensor; the green shaded areas indicate the AI-based partitioned swing phase (1–4 gait cycles); the dashed vertical red line indicates the 3D CGA-based identifications of foot-strike and toe-off events (red arrows from left to right).

**Table 1 sensors-22-04957-t001:** Mean and standard deviation (std) of gait parameters in the training dataset.

Condition	Step Length Left (Mean (m))	Step Length Right (Mean (m))	Step Length Left std (m)	Step Length Right std (m)	Gait Speed (m/s)	Number of Included Steps
Barefoot	0.37	0.389	0.052	0.038	0.876	1
Barefoot	0.465	0.473	0.013	0.03	1.019	1
Barefoot	0.282	0.444	0.037	0.013	0.389	2
Barefoot	0.457	0.417	0.012	0.02	0.797	1
Mean	0.394	0.431	0.029	0.025	0.770	
std	0.086	0.036	0.019	0.011	0.270	

**Table 2 sensors-22-04957-t002:** Mean and standard deviation (std) of gait parameters in the test dataset.

Condition	Step Length Left (Mean (m))	Step Length Right (Mean (m))	Step Length Left (std (m))	Step Length Right (std (m))	Gait Speed (m/s)	Number of Steps Left	Number of Steps Right
Barefoot	0.56	0.609	0.047	0.043	1.292	16	16
Barefoot	0.318	0.345	0.027	0.021	0.837	24	23
Shoes/Orthotics	0.341	0.36	0.052	0.041	0.87	7	7
Barefoot	0.471	0.496	0.032	0.044	1.053	23	21
Barefoot	0.451	0.452	0.036	0.037	1.028	20	20
Barefoot	0.298	0.315	0.037	0.045	0.823	23	22
Barefoot	0.306	0.457	0.022	0.016	0.399	5	4
Barefoot	0.507	0.501	0.049	0.013	1.256	19	19
Barefoot	0.446	0.423	0.022	0.016	0.814	25	22
Barefoot	0.538	0.502	0.026	0.063	1.153	24	23
Barefoot	0.389	0.439	0.043	0.025	0.771	21	20
Barefoot	0.573	0.604	0.023	0.017	0.998	20	18
Barefoot	0.511	0.521	0.05	0.03	1.092	23	21
Barefoot	0.336	0.341	0.036	0.024	0.845	32	30
Barefoot		0.446		0.015	0.699	9	10
Mean	0.432	0.454	0.036	0.030	0.929	19.4	18.4
std	0.099	0.089	0.011	0.015	0.231	7.3	6.7

**Table 3 sensors-22-04957-t003:** Mean and standard deviation (std) of gait parameters in the clinical validation study.

Condition	Step Length Left (Mean (m))	Step Length Right (Mean (m))	Step Length Left std (Mean (m))	Step Length Right std (m)	Gait Speed (Mean (m/s))	Number of Steps Left	Number of Steps Right
Barefoot	0.149	0.24	0.06	0.056	0.134	7	7
Barefoot	0.467	0.501	0.046	0.045	1.133	29	30
Shoes/Orthotics	0.439	0.485	0.053	0.053	0.979	43	41
Barefoot	0.254	0.032	0.014	0.01	0.184	27	21
Shoes/Orthotics	0.278	0.058	0.017	0.025	0.228	11	12
Barefoot	0.222	0.25	0.023	0.011	0.287	10	11
Shoes/Orthotics	0.284	0.362	0.024	0.022	0.449	8	8
Barefoot	0.475	0.14	0.024	0.022	0.237	24	22
Shoes/Orthotics	0.478	0.186	0.02	0.042	0.273	29	28
Mean	0.338	0.250	0.031	0.031	0.433	20.9	20
std	0.119	0.160	0.016	0.016	0.344	11.8	10.9

## Data Availability

The data presented in this study are available on request from the corresponding author. The data are not publicly available due to ethical and privacy reasons.

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
