# Peer review of "Flexible Machine Learning Algorithms for Clinical Gait Assessment Tools"

_sensors, 2022, doi:10.3390/s22134957_

Round 1

Reviewer 1 Report

[Comment 1] Novelty

I understood that the authors refer to the reinforcement learning (RL) method when stating flexible machine learning (ML) method. Is it true? (or is there any new factor incorporated in the flexibility of the method?). If the authors only refer to the RL, then they need to compare this study with previous study using RL with the same/similar research topic. I suggest the authors make a comparison table and clearly show which part is new in their study.

[Comment 2] Method

I understand that the authors might want to keep the methods limited somehow because of the copyright issue, but to allow the readers to gain any benefit from the study, the authors at least need to share the information about the research steps and algorithms used (related to the data analysis process) clearly. I expect to find something like a pseudocode or references to the used RL and DNN. The authors might need to address why such methods are used in contrast to other available methods as well. Even though initially I thought the research was very interesting, I could barely check or comment on anything because there are almost no details about the methods at all.

[Comment 3] Data and Results

[Subcomment 3a] I tried to open the supplementary file link but the file was not opened (I did not have any access). I suggest the authors include the figure into the appendix.

[Subcomment 3b] I think the RL usage is interesting and important in this study. When the authors stated that the RL could automatically adjust the system parameters to the user's condition, the authors need to present exact experiments that explain about that result, e.g., by showing the comparison when RL is used and not used, also, by showing intuitively how the RL works to adjust the parameters (especially using the experiment results).

[Subcomment 3c] I believe that because there are a lot of ML-related studies now, the authors need to start analyzing the applicability of the methods from the viewpoint of the practitioners. Please try to assess more about how practitioners really implemented similar ML-related study results, then discuss more on which parts of the proposed methodology are already ready to be implemented. Also, please address parts which are not ready yet for implementation at real medical systems, and how to make them ready.

[Comment 4] Writing quality and clarity

(Section 1) Please at least provide the definition of clinical gait analysis that would be useful especially for readers that are new to this topic.

Reviewer 2 Report

The authors are commended for their study and clear presentation.

Below are a few minor remarks which can possible improve the manuscript:

Line 20-2: Rephrase sentence for clarity.

Line 22: Unclear what the steps in brackets mean. Should the authors add the word "respectively" 

Line 55-60: Shorten for clarity.

Line 73: Consider adding a brief explanation for the terms "clinical" and "technical" flexibility.

Line 81: What is long short-term memory?

The introduction could be shortened by removing/condensing unnecessary information, e.g. lines 34-7, 55-60, 72-89, 90-110.

Line 127-9: Add model and version numbers. How many cameras?

Line 150, 179: Replace "custom made" by "custom".

Figure 1: Annotations are unclear. 

Line 224 & 244: Consider adding boxplots to illustrate the distribution on the errors.

Line 259: Instead of restating the aim, state the main finding of the study and its importance.

The discussion can be shortened.

Round 2

Reviewer 1 Report

Please revise some cut tables, e.g., by reducing the font size.

Author Response

We again thank the reviewers for their time to provide feedback on our manuscript.

Comment Reviewer 1: Please revise some cut tables, e.g., by reducing the font size

Author response: Thanks for pointing this out. We adjusted the size of Tables 1 - 3 to better match the manuscript format.